# T³OMVP: A Transformer-Based Time and Team Reinforcement Learning Scheme for Observation-Constrained Multi-Vehicle Pursuit in Urban Area

Zheng Yuan [ID], Tianhao Wu [ID], Qinwen Wang [ID], Yiying Yang, Lei Li *[ID] and Lin Zhang *

School of Artificial Intelligence, Beijing University of Posts and Telecommunications, 10 Xitucheng Road, Haidian Distinct, Beijing 100876, China; yuanzheng@bupt.edu.cn (Z.Y.); wu.tianhao@bupt.edu.cn (T.W.); wangqinwen@bupt.edu.cn (Q.W.); yyying@bupt.edu.cn (Y.Y.)
* Correspondence: leili@bupt.edu.cn (L.L.); zhanglin@bupt.edu.cn (L.Z.)

**Abstract:** Smart Internet of Vehicles (IoVs) combined with Artificial Intelligence (AI) will contribute to vehicle decision-making in the Intelligent Transportation System (ITS). Multi-vehicle pursuit (MVP) games, a multi-vehicle cooperative ability to capture mobile targets, are gradually becoming a hot research topic. Although there are some achievements in the field of MVP in the open space environment, the urban area brings complicated road structures and restricted moving spaces as challenges to the resolution of MVP games. We define an observation-constrained MVP (OMVP) problem in this paper and propose a transformer-based time and team reinforcement learning scheme (T³OMVP) to address the problem. First, a new multi-vehicle pursuit model is constructed based on Decentralized Partially Observed Markov Decision Processes (Dec-POMDPs) to instantiate this problem. Second, the QMIX is redefined to deal with the OMVP problem by leveraging the transformer-based observation sequence and combining the vehicle's observations to reduce the influence of constrained observations. Third, a simulated urban environment is built to verify the proposed scheme. Extensive experimental results demonstrate that the proposed T³OMVP scheme achieves improvements relative to the state-of-the-art QMIX approaches by 9.66~106.25%, from simple to difficult scenarios.

**Keywords:** multi-agent systems; multi-agent reinforcement learning; Internet of Vehicles; urban area

## 1. Introduction

The Internet of Vehicles (IoVs) is a typical application of Internet of Things (IoT) technologies in the Intelligent Transportation System (ITS) [1–3]. With the Vehicle-to-Vehicle (V2V) and Vehicle-to-Infrastructure (V2I) communications, the IoVs combined with Artificial Intelligence (AI) can improve the decision making of vehicles and the efficiency of ITS [4,5]. Reinforcement learning (RL) has been widely used in the ITS as a representative of advanced technology in AI [6]. Furthermore, multi-agent reinforcement learning (MARL) has been used in the IoVs to reduce communication latency and enhance communication efficiency [7,8]. The multi-vehicle pursuit (MVP) game describes a multi-vehicle cooperative ability to capture mobile targets, represented by the New York City Police Department guideline on the pursuit of suspicious vehicles [9]. The MVP game is becoming a hot topic in the IoVs supported by AI research.

There are mainly two ways to solve the MVP game: one is the game theory, and the other is cooperative MARL. Concerning the game theory, Eloy et al. proposed a team cooperative optimal solution for the border-defense differential game [10], and Huang et al. proposed a decentralized control scheme based on the Voronoi partition of the game domain [11]. However, it becomes challenging for game theory solutions to define a suitable objective function when the problem becomes more complex, such as the increased number of agents, restricted movement, and a complicated environment.



Concerning cooperative MARL research for the MVP game, the multi-agent system is modeled using Markov Decision Processes (MDP) [12], and a neural network can be used to approximate the complex objective function [13]. Cristino et al. used the Twin Delayed Deep Deterministic Policy Gradient (TD3) to demonstrate a real-world pursuit–evasion in the open environment with boundaries [14]. Timothy used the Deep Deterministic Policy Gradient (DDPG) with omnidirectional agents [15]. Thomy et al. proposed a Resilient Adversarial value Decomposition with the Antagonist-Ratios method and verified the method in the predator–prey scenario [16]. Jiang et al. proposed a vehicular end–edge–cloud computing framework to implement vertical and horizontal cooperation among vehicles [17]. Peng et al. proposed a Coordinated Policy Optimization to facilitate the cooperation of agents at both local and global levels [18]. However, the above studies in cooperative MARL are not verified in urban environments with complicated road structures, restricted moving spaces, and constrained observations due to architectural obstructions.

This paper introduces the transformer block to process the observation-entities of all agents inspired by [19]. The QMIX [20] is used as the baseline to control the pursuing vehicles, which is a state-of-the-art MARL algorithm that was successfully applied to other domains. Furthermore, this paper modifies the predator–prey scenario in [16] by adding intersections to the original open space to construct a multi-vehicle pursuing urban area for instantiating the MVP game. The proposed transformer-based time and team reinforcement learning scheme ($T^3$OMVP) is presented in Figure 1.

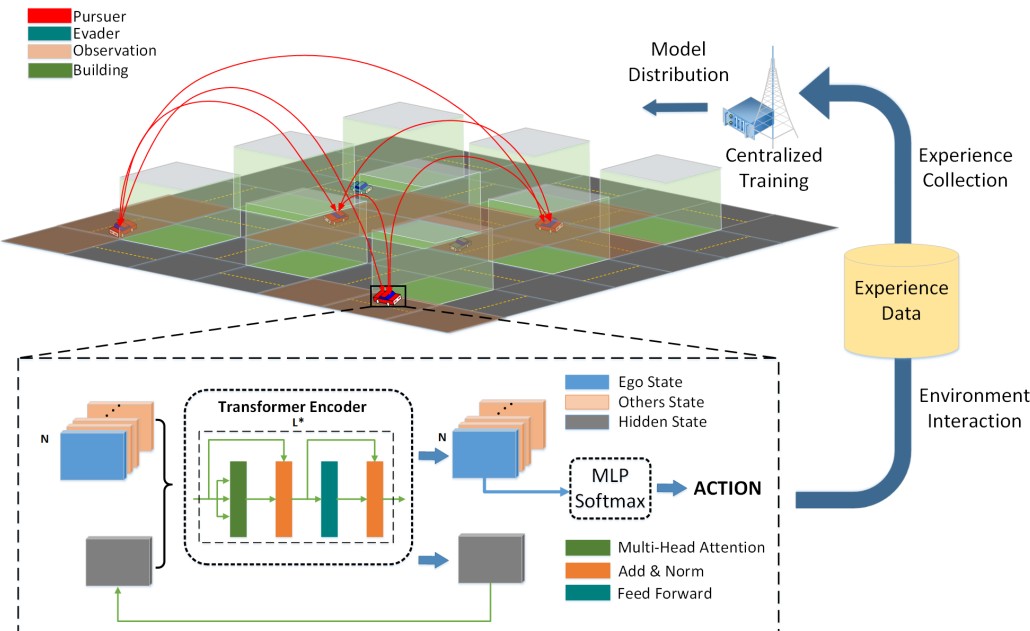

**Figure 1.** An overview of $T^3$OMVP for the problem of multi-vehicle pursuit in the multi-intersection urban area. The transformer encoder is used to process the observation-entities of all pursuers. The hidden state is used to store historical observations.

The main contributions of this paper are:

1. The MVP game in the urban environment is defined as the observation-constrained multi-vehicle pursuit (OMVP) problem with the occlusion of buildings to each pursuing vehicle.
2. The $T^3$OMVP scheme is proposed to address the OMVP problem using Dec-POMDPs and reinforcement learning. The $T^3$OMVP introduces the transformer block to deal with the observation sequences to realize the multi-vehicle cooperation without a policy decoupling strategy.
3. A novel low-cost and light-weight MVP simulation environment is constructed to verify the performance of the $T^3$OMVP scheme. Compared with large-scale game environments,

this simulation environment can improve the training speed of reinforcement learning. All source codes are provided on GitHub (https://github.com/its-ant-bupt/T3OMVP) (last accessed on 27 February 2022).

The details of the MVP problem and its relationship with Dec-POMDPs are presented in Section 2. The OMVP problem and the specific implementation details of the T$^3$OMVP scheme are presented in Section 3. The experimental results and discussions are provided in Section 4. Section 5 concludes this paper.

## 2. Multi-Vehicle Pursuit

### 2.1. Problem Statement

This paper mainly describes a multi-intersection urban area, as shown in Figure 1. In a $W \times W$ grid, there are $N$ learning red pursuing vehicles and $\frac{N}{2}$ randomly moving blue evading vehicles. The pursuing vehicle has an observation field of $M \times M$, but the shape of its observation field is a cross or a straight line corresponding to the scenarios of an intersection or a straight road due to the influence of obstacles. According to the description in [9], each pursuing vehicle will share its observation and position with other pursuing vehicles. The shared observation is the position of the evading vehicle in the field of vision. In addition, each vehicle also has an observation field showing the position of the obstacles in its $M \times M$ field. Therefore, the *state* of each pursuing vehicle includes its own $M \times M$ observation and those shared by other vehicles. The *global state* is a $W \times W$ observation. The *reward* of each pursuing vehicle is calculated by the normalized function $g = \frac{1}{n}$. If there are $n \in \{1, \dots, N\}$ pursuing vehicles capturing the same evading vehicle, the *reward* of each corresponding pursuing vehicle is $\frac{1}{n}$. The *global reward* is the sum of all the pursuing vehicles' rewards, and an evading vehicle is captured with the *global reward* of +1.

### 2.2. Dec-POMDP

In the scenario of the urban area, each vehicle can be modeled as an agent which can only have partial observation of the environment. As such, the cooperative multi-agent MVP task characterized by communications between agents can be formulated as a partially observable Markov game $G = \langle S, K, A, P, r, O, Z, n, \gamma \rangle$. The *global state* of the environment is denoted by $s \in S$. At each step $t$, each agent $k \in \mathbf{K} \equiv \{1, \dots, N\}$ chooses an action $a^k \in A$, forming a joint action $\mathbf{a} \in \mathbf{A} \equiv A^n$. Thus, the state transition function $P(s' \mid s, \mathbf{a}) : S \times \mathbf{A} \times S \to [0, 1]$ represents a transition in the environment. A *global reward* function $r(s, \mathbf{a}) : S \times \mathbf{A} \to R$ is necessary in MARL to estimate a policy; this paper shares the same reward function among agents. Meanwhile, a partially observable scenario is considered in which each agent draws an individual observation $o \in O$ according to observation function $Z(s, k) : S \times K \to O$. Each agent has an action-observation history $\tau^k \in T \equiv (O \times K)^*$ on which it conditions a stochastic policy $\pi^k \left( a^k \mid \tau^k \right) : T \times A \to [0, 1]$. The joint policy $\pi$ has a joint action-value function $Q^\pi(s_t, \mathbf{a}_t) = \mathbb{E}_{s_{t+1:\infty}, \mathbf{a}_{t+1:\infty}}[R_t \mid s_t, \mathbf{a}_t]$, where $R_t = \sum_{i=0}^{\infty} \gamma^i r_{t+i}$ is a discounted reward and $\gamma \in [0, 1)$ is the discount factor.

## 3. T$^3$OMVP Scheme

The OMVP problem can be formulated by cooperative MARL, and each pursuing vehicle can be modeled as an agent with state, action, and reward.

### 3.1. State, Action, and Reward

According to the assumptions of the MVP problem, each pursuing vehicle can obtain the observations of other pursuing vehicles through V2V or V2I communication. At time $t$, $p_{\text{pur,k}}^t = (i_k^t, j_k^t)$ represents the position of the $k$th pursuer, $p_{\text{eva,m}}^t = (i_m^t, j_m^t)$ represents the position of the $m$th evader, and $p_{\text{obs,n}} = (i_n, j_n)$ represents the position of the $n$th obstacle area. Therefore, $P_{\text{pur}}^t = (p_{\text{pur,1}}^t, \dots, p_{\text{pur,N}}^t)$ means the positions of all pursuers, $P_{\text{eva}}^t = (p_{\text{eva,1}}^t, \dots, p_{\text{eva,N/2}}^t)$ means the positions of all evaders, and $P_{\text{obs}} = (p_{\text{obs,1}}, \dots)$ means the positions of all obstacle

areas. It is defined that the observation of the $k$th pursuer at time $t$ is divided into two parts: evading target observation and obstacle observation, which are represented by $E_k^t$ and $B_k^t$, respectively. Both $E_k^t$ and $B_k^t$ are represented by an $M \times M$ matrix, where $E_k^t$ is limited by urban roads and $B_k^t$ is not limited. $e_{k,i,j}^t$ and $b_{k,i,j}^t$ are the elements in row $i$ and column $j$ in $E_k^t$ and $B_k^t$, respectively. Define $o_k^t \in E_k^t$ as the observable area of the $k$th pursuer, whose shape is a cross or a straight line. For the OMVP problem in grid environment, let

$$
\begin{aligned}
e_{k,i,j}^t &= \begin{cases} 1 & \text{if } (i,j) \in o_k^t \text{ and } (i,j) \in P_{\text{eva}}^t \\ 0 & \text{others} \end{cases} \\
b_{k,i,j}^t &= \begin{cases} 1 & \text{if } (i,j) \in P_{\text{obs}} \\ 0 & \text{others} \end{cases}.
\end{aligned}
\tag{1}
$$

In order to achieve vehicle cooperative pursuit, the *state* of each pursuing vehicle should include the observations of all other vehicles. $s_i^t = (E_i^t, B_i^t, E_{\text{other}}^t)$ represents the *state* of $i$th pursuer at the time $t$, including the evading vehicle observation, the obstacle observation of $i$th pursuer, and the evading vehicle observation of the other pursuers.

Because the urban area is built in the grid world, the action space of pursuing vehicles can be divided into five parts, including moving forward, moving backward, turning left at the intersection, turning right at the intersection, and stopping. The *reward* and *global reward* in the OMVP are the same as in the MVP.

### 3.2. Centralized Training with Decentralized Execution

This paper considers that extra state information is available and vehicles can communicate freely. Centralized training with decentralized execution (CTDE) is a standard paradigm in MARL. In the training process, the centralized value function, which conditions on the global state and the joint actions, is obtained. Meanwhile, each agent utilizes individual action-observation histories to learn its individual value function, which is updated by a centralized gradient provided by the centralized value function. In the executing process, the learnt policy for each agent conditioning on the individual value function can be executed in a decentralized way. State-of-the-art MARL algorithms, such as the VDN or QMIX, adopt this architecture.

### 3.3. Observation-Constrained Multi-Vehicle Pursuit

The OMVP can be modeled as a Dec-POMDP. This paper uses the QMIX to deal with the multi-agent credit assignment in the MVP. At the same time, in order to deal with the problem that the observation area is affected by the complex environment, the transformer is used to process the time observations and team observations.

#### 3.3.1. Monotonic Value Function Factorization

Deep Q-Networks (DQNs) contain the experience replay mechanism and the target network to fit the action-value function better, which can be used in MARL. The cost function of DQNs is the standard squared *TD error*:

$$
L(\theta) = E_{s,a,r,s'}\left[(y_i^{DQN} - Q(s, \mathbf{a} \mid \theta))^2\right],
\tag{2}
$$

where $a$ is the action at the current state $s$, $s'$ is the next state, $y^{DQN} = r + \gamma \max_{\mathbf{a'}} \overline{Q}(s', \mathbf{a'} \mid \bar{\theta})$ and $r$ is the received reward. $\bar{\theta}$ are the parameters of *target network* $\overline{Q}$ which are periodically copied from the $\theta$ of *current network* $Q$.

At present, cooperative MARL often uses CTDE for training but requires a centralized action-value function $Q_{\text{total}}$ that conditions on the global state and the joint action. $Q_{\text{total}}$ can usually be approximated using a neural network by $\hat{Q}_{\text{total}}$, which is factorized to approximate individual $\hat{Q}_i$ for each agent $i$ in order to update $\hat{Q}_i$. The objective of the update is to minimize the following loss which is analogous to the standard DQN loss:

$$L(\theta) = \sum_{i=1}^{b} \left[ \left( y_i^{total} - \hat{Q}_{total}(\boldsymbol{\tau}, \mathbf{a}, s; \theta) \right)^2 \right], \tag{3}$$

where $b$ is the batch size to sample from the replay buffer, $y^{total} = r + \gamma \max_{\mathbf{a}'} \overline{\hat{Q}}_{total}(\boldsymbol{\tau}', \mathbf{a}', s'; \bar{\theta})$.

VDN is the simplest method to approximate the individual action-value function. It formulates $\hat{Q}_{\text{total}}$ as a sum of individual action-value functions $\hat{Q}_i(\tau_t^i, a_t^i)$, one for each agent $i$, which condition only on individual action-observation histories:

$$\hat{Q}_{\text{total}}(\boldsymbol{\tau}, \mathbf{a}) = \sum_{i=1}^{N} \hat{Q}_i(\tau^i, a^i; \theta^i), \tag{4}$$

where $\boldsymbol{\tau} \in \mathbf{T} \equiv T^N$ is the action-observation history of all agents, $\mathbf{a} \in \mathbf{A}$ is the joint action, $\tau_t^i \in T$ and $a_t^i \in A$ is the action-observation history and action of agent $i$, respectively.

Because the full factorization of VDN is not necessary to extract decentralized policies that are fully consistent with their centralized counterpart, the QMIX uses a mixing network to represent the $\hat{Q}_{\text{total}}$. The mixing network is a feed-forward neural network that takes the agent network outputs as input and mixes them monotonically. The weights of the mixing network are restricted to be non-negative to enforce the monotonicity constraint of Equation (5).

$$\frac{\partial \hat{Q}_{\text{total}}}{\partial \hat{Q}_i} \geq 0, \forall i \in \mathbf{K} \tag{5}$$

3.3.2. Observations Sequence

For the OMVP problem, although the state acquired by the pursuing vehicle is incomplete, more historical observations endow pursuing vehicles with better decisions of the next action. As such, the MVP problem becomes a Dec-POMDP. A coordinated method of time observations and team observations, named *TT-Observations*, is used to process the joint observation of all pursuing vehicles, thus solving the problem of constrained observations limited by the complex urban environment.

Time Observations

Because pursuing vehicles can save part of the road information in the historical observation, a more extensive observation range can be obtained by combining multiple historical observations. Therefore, when other pursuing vehicles find the position of the target vehicle in the historically observed road, the pursuing vehicle can trace the source, thereby improving the pursuing efficiency. For example, in Figure 2a, the orange area is the historical time observations of the red pursuing vehicle No. 1. When it arrives at the position $P$, an evading vehicle appears two blocks behind it. Due to the limitation of its field of view, pursuing vehicle No. 1 cannot observe the evading vehicle. Although the red pursuing vehicle No. 2 can observe the evading vehicle, it still cannot pursue the evading vehicle after turning around, due to the evading vehicle being able to move to the right. Because the red pursuing vehicle No. 1 can drive in the opposite direction, it can trace the source based on historical time observations to capture the evading vehicle. Furthermore, the previously observed information is added to the input of the transformer block to process time observations using the self-attention mechanism.

Team Observations

A more comprehensive observation range can also be constructed through integrating the observations of all pursuing vehicles. In the IoVs, vehicles can communicate through V2V or V2I, so the team's observation sequences can be used to deal with complicated roads in urban environments. When multiple pursuing vehicles observe the same target, it is possible to avoid all pursuing vehicles aiming at the same target by focusing on the observation sequence of all vehicles, so that the target can be dispersed and the pursuing efficiency can be improved. For example, in Figure 2b, the yellow area is observations of all

the red pursuing vehicles. When two pursuing vehicles observe the blue evading vehicle No. 2 at the same time, by processing all observation sequences, it is possible to avoid two red vehicles pursuing the blue evading vehicle No. 2 at the same time, and pursue the most suitable vehicle separately, thereby improving the pursuing efficiency. Furthermore, the joint observation of all pursuing vehicles is used as the input of the transformer block to process team observations using the self-attention mechanism.

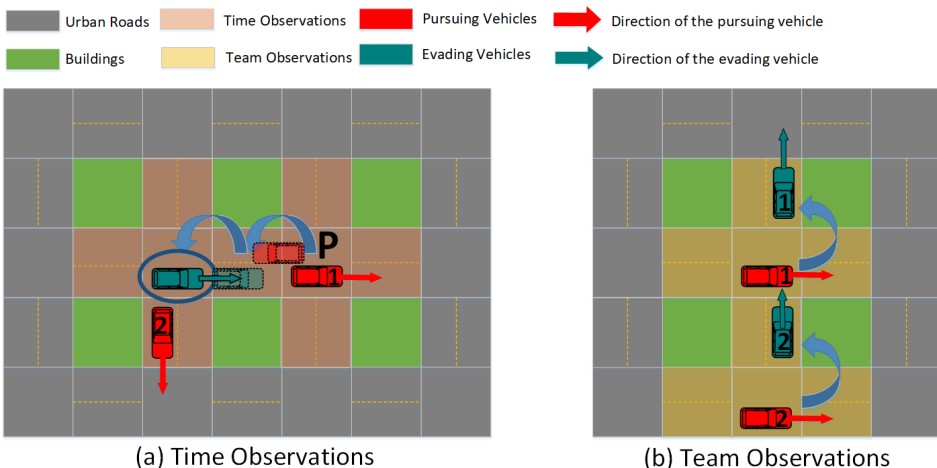

**Figure 2.** (**a**) Time observations are used to backtrack historical paths: pursuing vehicle No. 1 traces the previous location of the evading vehicle in the time observation of pursuing vehicle No. 2 and turns around to capture the evading vehicle. (**b**) Team observations are used to assign team goals: pursuing vehicle No. 1 chooses to capture evading vehicle No. 1 instead of evading vehicle No. 2 which is in the observation of pursuing vehicle No. 2.

*TT-Observations*

There are two ways to combine time observations and team observations. One is that the team observations are the main body, and the time observations are used as an additional input. The other is that the time observations are the main body, and the team observations are used as an additional input. In the recurrent neural network, time information is usually transmitted and encoded through the hidden layer. Using the transformer structure based on team observations, the self-attention mechanism can be fully utilized to realize the cooperative control of the team. If the transformer structure based on time observations is used, the self-attention will only be placed on a single pursuing vehicle and cannot realize the cooperative control of the team. Therefore, the $T^3$OMVP adopts the transformer structure based on team observations and uses a hidden layer $h$ to store time observations as an additional input.

### 3.3.3. Transformer-Based *TT-Observations*

This paper uses the method *TT-Observations* that combines time observations and team observations and uses the transformer block to calculate the *Q*-value of each agent through the self-attention mechanism based on UPDeT.

Self-Attention Mechanism

Vaswani et al. first used the self-attention mechanism in the transformer block [21]. In the self-attention, each input embedding vector *X* has three different vectors, *Q*, *K*, and *V*, representing query, key, and value, respectively. *Q*, *K*, *V* are obtained by multiplying three different weight matrices $W_q$, $W_k$, $W_v$ with the embedding vector *X*, and the dimensions of the three weight matrices are the same. The calculation formula of self-attention output is as follows:

$$\text{Attention}\,(Q, K, V) = \text{softmax}\left(\frac{QK^T}{\sqrt{d_k}}\right)V, \tag{6}$$

where $d_k$ equals the number of columns of the $Q$ and $K$ matrices to prevent the inner product from being too large; $K^T$ represents the transpose of the matrix $K$.

Transformer in Time and Team Observations

This paper assumes that the observation can be transmitted among the pursuing vehicles through communication. At time t, the observation set of all pursuing vehicles is $O^t$, which contains independent observations of $N$ agents $\{o_1^t, o_2^t, \ldots, o_N^t\}$. The independent observations of all pursuing vehicles can be encoded into embedding vectors of the same length through the embedding layer, so the embedding vector of the entire team observations can be obtained as follows:

$$E_t = \{\mathrm{Emb}(o_1^t), \mathrm{Emb}(o_2^t), \ldots, \mathrm{Emb}(o_N^t)\}, \tag{7}$$

where Emb represents the embedding layer in the neural network. Unlike UPDeT, this paper uses the joint observation of all pursuing vehicles as the input of the transformer block to realize the overall decision-making control of the pursuing vehicles. Similar to UPDeT, this paper uses a hidden state $h_{t-1}$ to record the historical observation state to achieve historical backtracking. The observed embedding vector and the hidden state are combined at time $t-1$ as the input of the transformer block to obtain $X_1$, and $X_l$ ($l \in \{1, 2, \ldots, L\}$) is the input of the transformer block of the $l$th layer; $L$ is the number of layers in the transformer encoder. The whole calculation process is as follows:

$$\begin{aligned} X_1 &= \{E_t, h_{t-1}\} \\ Q_l, K_l, V_l &= \mathrm{LF}_{Q,K,V}(X_l) \\ X_{l+1} &= \mathrm{Attention}\,(Q_l, K_l, V_l). \end{aligned} \tag{8}$$

where LF represents the linear functions used to compute $Q_l$, $K_l$, and $V_l$. The T$^3$OMVP scheme further uses the output of the last self-attention layer as the input of the linear layer LN that calculates $Q$-value:

$$Q^t(E_t, h_{t-1}, \mathbf{a}) = \mathrm{LN}(X_{l+1}, \mathbf{a}) = \{Q_1^t, Q_2^t, \ldots, Q_N^t\}, \tag{9}$$

where $Q_i^t (i \in \{1, 2, \ldots, N\})$ represents the $Q$-value of each agent at time $t$, and $Q^t$ represents the set of $Q$-values of all agents. After obtaining the $Q$-values of all agents, the global $Q$-value can be calculated by credit assignment function:

$$Q_{\mathrm{total}}^t(\boldsymbol{\tau}_t, \mathbf{a}_t) = \mathrm{F}(Q_1^t, Q_2^t, \ldots, Q_N^t) = \mathrm{F}(Q^t(E_t, h_{t-1}, \mathbf{a})), \tag{10}$$

where F is the credit assignment function. In this paper, the QMIX is used to calculate the global $Q$-value. In the QMIX, F is a mixed linear network to ensure that the derivative of the global $Q$-value and the individual $Q$-value is positive.

Decision-Making and Training Process

In UPDeT, in order to make the QMIX added with the transformer block more effective than the original QMIX, UPDeT adopts a policy decoupling strategy, that is, all action groups are calculated separately for the $Q$-value of each agent, which will make the calculation more complicated and difficult to apply to other scenarios.

In the T$^3$OMVP scheme, the local information observed by each agent is jointed as shown in Figure 3. After each pursuing vehicle obtains the observations of other pursuers through V2V or V2I, all the information will be jointed linearly. On the one hand, the action set in the OMVP problem is small, and it is inconvenient to divide the action set into multiple action pairs. On the other hand, the larger-range joint observation can include the set of targets observed by all pursuing vehicles to facilitate cooperative control. Therefore, the T$^3$OMVP scheme does not have to adopt the policy decoupling strategy. The following experiments show that, in the T$^3$OMVP scheme, jointing the observation into

the network without using policy decoupling can achieve the same benefits as adopting policy decoupling.

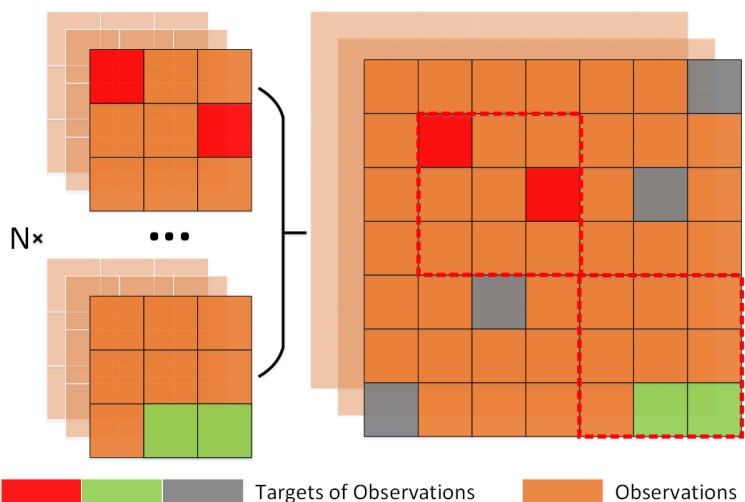

**Figure 3.** Generating joint observation. The observations of N pursuing vehicles are jointed into a larger observation matrix linearly.

The training process of the T³OMVP scheme is shown in Algorithm 1. At the beginning of each episode, the joint observation $\mathbf{o}_0$ of each agent can be obtained from the global initial state $s_0$. In each step of simulating, actions are generated by the $\epsilon$-greedy strategy; note that the $\epsilon$-greedy strategy will be defined in Section 4.2. After obtaining the next observation and reward $(\mathbf{o}_{t+1}, r_t)$, $(\boldsymbol{\tau}_t, \mathbf{a}_t, r_t, \boldsymbol{\tau}_{t+1})$ is stored to $M$. Finally, a minibatch of transitions are sampled from $M$ to update all networks.

---

**Algorithm 1:** T³OMVP On-line training

---

**1** Initialize replay memory $M$;

**2** Initialize individual value network $\hat{Q}_i$ and global value network $\hat{Q}_{\text{total}}$ with random parameters $\theta_i$ and $\theta$;

**3** Initialize target individual value parameters $\bar{\theta}_i = \theta_i$ and target global value parameters $\bar{\theta} = \theta$;

**4** **for** $episode = 1$ **to** $E$ **do**

**5**     Initialize hidden state $h_0$;

**6**     Observe initial state $s_0$ and joint observation $\mathbf{o}_0 = [Z(s_0, i)]_{i=1}^N$;

**7**     **for** $t = 1$ **to** $T$ **do**

**8**         With probability $\epsilon$ select a random action $a_t^i$;

**9**         Otherwise $a_t^i = \arg\max_{a_t^i} Q_i(\tau^i, a^i, h_{t-1}; \theta^i)$;

**10**         Take action $\mathbf{a}_t$, and get the next observation and reward $(\mathbf{o}_{t+1}, r_t)$;

**11**         Store transition $(\boldsymbol{\tau}_t, \mathbf{a}_t, r_t, \boldsymbol{\tau}_{t+1})$ in $M$;

**12**         Sample a random minibatch of transitions $(\boldsymbol{\tau}, \mathbf{a}, r, \boldsymbol{\tau}')$ from $M$;

**13**         Set $y^{total}(r, \boldsymbol{\tau}'; \bar{\theta}) = r + \gamma \overline{\hat{Q}}_{total}(\boldsymbol{\tau}', \mathbf{a}', s'; \bar{\theta})$,

                $\mathbf{a}' = \left[\arg\max_{a^i} \hat{Q}_i(\tau^{i,\prime}, a^i, h'; \bar{\boldsymbol{\theta}})\right]_{i=1}^N$;

**14**         Update the $\hat{Q}_i$ and $\hat{Q}_{\text{total}}$ by minimizing the loss:

                $L = \sum_{i=1}^b \left[\left(y_i^{total} - \hat{Q}_{total}(\boldsymbol{\tau}, \mathbf{a}, s; \theta)\right)^2\right]$;

**15**         Update the target networks parameters $\hat{\theta}_i = \theta_i$ and $\hat{\theta} = \theta$ with period $I$.

**16**     **end**

**17** **end**

---

## 4. Results

### 4.1. Evader Strategy

In this paper, four movement strategies are designed for evaders to test the robustness of the T$^3$OMVP in the pursuit. The four strategies are: keeping still, moving around in the latitudinal direction, moving around in the longitudinal direction, and moving in a circle, as shown in Figure 4. In each episode of training and testing, the evading vehicle randomly selects and executes one strategy from the above four.

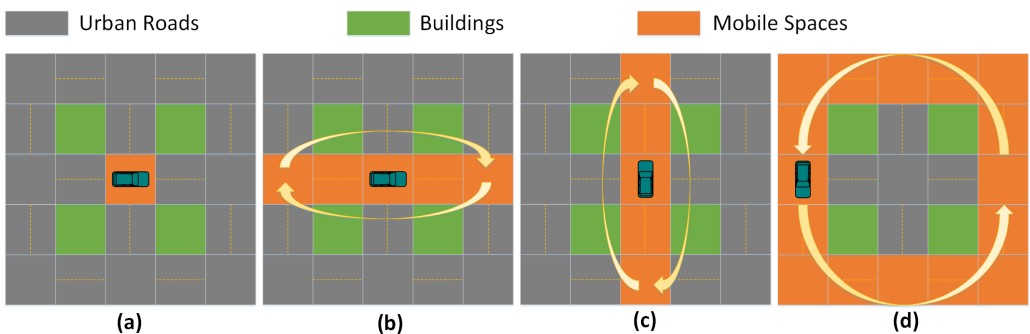

**Figure 4.** Evader Strategy. (**a**) Keeping still. (**b**) Moving around in the latitudinal direction. (**c**) Moving around in the longitudinal direction. (**d**) Moving in a circle.

### 4.2. $\epsilon$-Greedy

The T$^3$OMVP utilizes the $\epsilon$-greedy strategy when the agent makes a decision. Specifically, the agent randomly chooses an unknown action with probability $\epsilon$ to explore or exploits with probability $1-\epsilon$ by selecting the action with the largest $Q$-value among the existing actions. The $\epsilon$-greedy algorithm is as follows:

$$A^* = \arg\max_a Q(s, a)$$

$$\pi(a \mid s) = \begin{cases} 1 - \epsilon + \epsilon/|\mathcal{A}(s)| & \text{if } a = A^* \\ \epsilon/|\mathcal{A}(s)| & \text{if } a \neq A^* \end{cases}. \tag{11}$$

where $A^*$ represents the local optimal action obtained according to the $Q$-value.

### 4.3. Simulation Settings

The T$^3$OMVP scheme is trained and evaluated in a self-designed urban multi-intersection environment, developed with Python 3.6 (Python Software Foundation, Beaverton, OR, USA). The environment is based on the gym and contains multiple intersections in a $W \times W$ grid environment. In the experiment, $\lambda$ represents the ratio of pursuing vehicles to evading vehicles. In order to verify the effect of $\lambda$ on the stability of the T$^3$OMVP scheme, $\lambda$ is set variably with 2, 1, 0.5. Experimental scenarios include the 8v4 (eight pursuing vehicles vs. four evading vehicles) scenario, 4v4 (four pursuing vehicles vs. four evading vehicles) scenario, and 2v4 (two pursuing vehicles vs. four evading vehicles) scenario. The parameters of the relevant experimental environment are shown in Table 1.

This paper adopts QMIX as the baseline method, including two hypernetworks, and uses the *Elu* activation function after each layer of the network $DN(o)$ where $o$ represents the dimension of the network output. The *Elu* can avoid the disappearance of the gradient, reduce the training time, and improve the accuracy in the neural network [22]. Similar to UPDeT, after obtaining the joint observation, the T$^3$OMVP feeds it into the transformer together with the hidden state. The transformer encoder consists of multiple blocks, each of which contains a multi-head self-attention, a feed-forward neural network, and two resnet structures. In UPDeT, the QMIX with the transformer needs to adopt the policy decoupling strategy to achieve better performance than the QMIX with the transformer. However, the T$^3$OMVP adopts an extensive observation matrix integrating observations

of all pursuing vehicles to replace the policy decoupling strategy and achieves a better or same performance than UPDeT. The complete hyperparameters are listed in Table 2.

**Table 1.** Experimental Parameters.

| Parameters | Value |
|---|---|
| Number of episodes $E$ | 40,000 |
| Time step $T$ | 50 |
| Grid space width $W$ | $[13, 17, 21]$ |
| Intersection interval | 1 |
| Number of intersections | $[36, 64, 100]$ |
| Distance of vehicle moves each time | 1 |
| Size of the observation space of the pursuing vehicle | $5 \times 5$ |
| Size of the joint observation space of the pursuing vehicle | $[3 \times 13 \times 13, 3 \times 17 \times 17, 3 \times 21 \times 21]$ |
| Number of evading vehicles | 4 |
| Historical observation length | 5 |

**Table 2.** Hyperparameters for Neural Networks.

| Hyperparameters | Value |
|---|---|
| Batch size | 32 |
| Memory capacity $M$ | 20,000 |
| Learning rate | $0.001 \rightarrow 0$ |
| Optimizer | Adam |
| Discounted factor $\gamma$ | 0.95 |
| $\epsilon$ decay | 0.0001 |
| $\epsilon$ min | 0.1 |
| Period of update $I$ | 4000 |
| *QMIX* | |
| Hypernetwork w #1 | $[DN(128), Elu, DN(128 + N)]$ |
| Hypernetwork w #final | $[DN(128), Elu, DN(128)]$ |
| Hypernetwork b #1 | $[DN(128)]$ |
| Output | $[DN(128), Elu, DN(1)]$ |
| *Transformer Encoder* | |
| Transformer depth | 2 |
| Embedding vector length | 250 |
| Number of heads | 5 |

*4.4. Discussion*

The entire experiment is divided into three parts. In the first part, to compare the T$^3$OMVP with UPDeT, the unified reward function is used to evaluate the T$^3$OMVP, QMIX, QMIX + UPDeT, VDN, VDN + UPDeT, and T$^3$VDN, except M3DDPG which has been proven not to perform well in the pursuit–evasion scenario [16]. In this part, all methods are trained in the $13 \times 13$ multi-vehicle pursuit grid environment with the 8v4 scenario, 4v4 scenario, and 2v4 scenario, respectively. There are four strategies for evading vehicles as shown in Section 4.1. At test time, 50 episodes of verification are performed to calculate the average value of the reward as the reward for the current training step. In the second part, the performances of T$^3$OMVP, QMIX, and VDN are verified on multiple scenarios of different difficulty, including $\lambda \in \{2, 1, 0, 5\}$ and $W \in \{13, 17, 21\}$. Finally, to evaluate the role of the self-attention mechanism in decision making, the attention is analyzed in two aspects: team observations and time observations.

Figure 5 depicts the performance comparison of T$^3$OMVP, QMIX, QMIX + UPDeT, VDN, VDN + UPDeT, and T$^3$VDN in the $13 \times 13$ grid environment on the 8v4 scenario, 4v4 scenario, and 2v4 scenario, respectively. As shown in Figure 5a–c, the T$^3$OMVP scheme, based on QMIX and using joint observation rather than a policy decoupling strategy, can achieve the best performance on both the simple 8v4 scenario and the difficult

2v4 scenario. However, its performance on the 4v4 scenario is not as good as that of QMIX+UPDeT which uses the policy decoupling strategy. The reason is that on the 4v4 scenario, the observations dimension is close to the number of actions, so the action–observation pairs are more suitable for policy decoupling. However, when the observations dimension and the number of actions are different, policy decoupling will make it difficult to assign appropriate actions, resulting in a performance degradation even worse than the original QMIX in the 2v4 scenario. Furthermore, the T³OMVP weakens the influence of the observations dimension and number of actions, because the transformer combined with *TT-observations* endows the pursuing vehicle with more comprehensive information from all other vehicles, which avoids vehicles from falling into the local optimum situation, and thereby the pursuing efficiency can be improved in more scenarios. The purple line and dark blue line in Figure 5 indicate that VDN + UPDeT performs poorly in the OMVP scenario. This phenomenon can be interpreted as follows: On the one hand, the relatively simple reward mechanism in our scenario is more adaptive with a concise way of calculating the global *Q*-value; hence, the VDN can solely achieve relatively competitive performance. On the other hand, the update process involved in the VDN is relatively simple; thus, the VDN + UPDeT is not sufficient for significantly improving the original performance in the OMVP scenario. Given the result that the VDN combined with a transformer structure and *TT-observations* can still achieve better performance than the VDN, it proves that the transformer combined with *TT-observations* can improve or maintain the performance without the policy decoupling strategy.

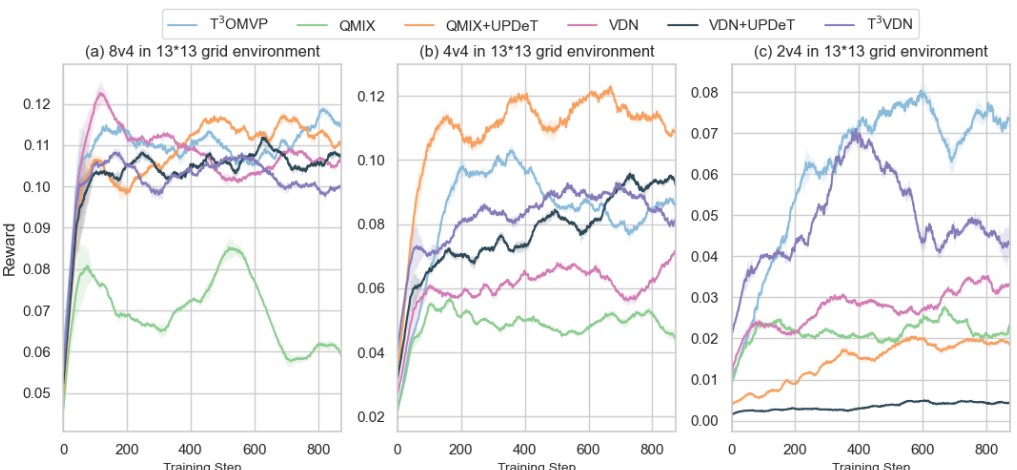

**Figure 5.** Comparison of T³OMVP and UPDeT methods in the 13 ∗ 13 grid environment in 8v4 scenario (**a**), 4v4 scenario (**b**), and 2v4 scenario (**c**), respectively. QMIX + UPDeT and VDN + UPDeT refer to methods using the UPDeT with policy decoupling strategy. T³VDN refers to VDN using transformer-based time and team observations without policy decoupling strategy.

Figure 6 shows the performances of the T³OMVP, QMIX, and VDN at different difficulty levels. According to the change of $\lambda$, Figure 6a–c show the situation in a $13 \times 13$ grid environment in the 8v4 scenario, 4v4 scenario, and 2v4 scenario, respectively. As shown in Figure 6a–c, in the simple 8v4 scenario, the VDN shows better performances than the QMIX and can even reach the performance of the T³OMVP. It indicates that the VDN can adapt to such situations better than the QMIX in relatively simple scenarios. As the number of pursuing vehicles decreases, the problem becomes complicated. The performance gap between the T³OMVP and QMIX gradually widens, and so does that between the T³OMVP and VDN. Furthermore, the performance gap between the VDN and QMIX gradually narrows. The results indicate that as the difficulty increases, the VDN with a simple structure cannot adapt to these scenarios anymore; thus, the gap between it and the QMIX gradually narrows, but the T³OMVP can still achieve the best performance. According to the change of *W*, Figure 6d,g show the situation in the 8v4 scenario in the $17 \times 17$ grid environment

and $21 \times 21$ grid environment, respectively. As shown in Figure 6a,d,g, it can be seen that as the size of the environment increases, the difficulty of pursuing increases, the VDN still gradually reaches the same performance as the QMIX, and the T³OMVP gradually shows better performance than the QMIX and VDN. It is consistent with the previous statement that the T³OMVP can show better performance in more difficult scenarios, because the T³OMVP can more comprehensively evaluate the observed information through the self-attention mechanism. At the same time, as the difficulty of the scenario increases, the gap between the QMIX and VDN gradually narrows, and the QMIX shows more competitive performance than the VDN in the 2v4 scenario in the $21 \times 21$ grid environment. It can be concluded that, as the difficulty increases, the QMIX can adapt to more complex scenarios, which is beneficial to the decision making among multi-vehicles. Furthermore, this paper compared the performance of the T³OMVP and QMIX on multiple test scenarios. In order to eliminate the influence of the evading vehicle movement, the strategy of the evading vehicles is keeping still, and the random number seed is fixed. The results are shown in Table 3. It is observed that transformer structure improves the performance of the QMIX by 9.66~106.25% on multiple test scenarios. For that reason, the T³OMVP scheme is generated by adding the transformer structure on the basis of the QMIX to deal with the OMVP problem in the urban multi-intersection environment.

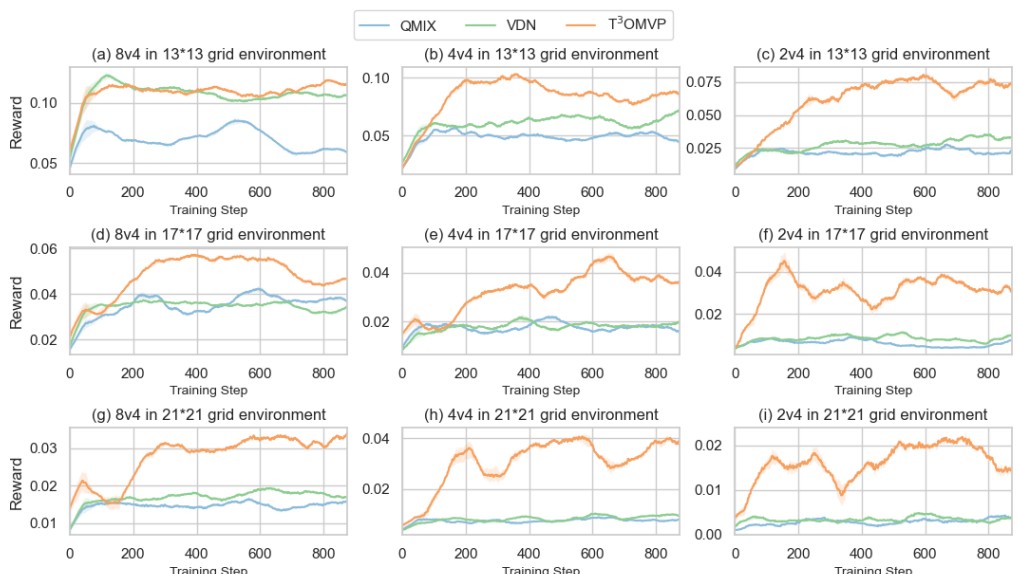

**Figure 6.** Comparison of T³OMVP, QMIX, and VDN in 8v4 scenario (**a,d,g**), 4v4 scenario (**b,e,h**), and 2v4 scenario (**c,f,i**) with $W \in \{13, 17, 21\}$.

Figure 7a describes the 8v4 scenario in the $13 * 13$ urban multi-intersection environment. As shown in Figure 7b, when two pursuing vehicles observe different targets, the attention of the pursuing vehicle No. 5 is the highest and that of the pursuing vehicle No. 7 ranks second. The reason is pursuing vehicles No. 5 and No. 7 are the closest vehicles to the evading vehicle and the calculated attention is also biased toward No. 5 and No. 7. In addition, Figure 7b also shows that the attention of the pursuing vehicle No. 2 is the highest except for the pursuing vehicles No. 5 and No. 7. The reason is that pursuing vehicle No. 2 is most relevant to the direction of the evading vehicle observed by No. 7. Therefore, focusing on pursuing vehicle No. 2 can improve the pursuing efficiency by mobilizing the No. 2 vehicle to pursue the evading vehicle observed by No. 7 in the subsequent actions.

**Table 3.** Testing for comparison of performance in 8v4 scenario, 4v4 scenario, and 2v4 scenario with $W \in \{13, 17, 21\}$. The bold texts are to represent the min and max to emphasize our outstanding performance.

| Sum of Rewards | Grid Space Width | 13 | 17 | 21 |
|---|---|---|---|---|
| | QMIX | 3.88 | 1.1775 | 0.76 |
| 8v4 scenario | T³OMVP | 4.255 | 1.5175 | 0.9175 |
| | Improvement | **9.66%** | 28.87% | 20.72% |
| | QMIX | 1.345 | 0.455 | 0.345 |
| 4v4 scenario | T³OMVP | 2.115 | 0.88 | 0.475 |
| | Improvement | 57.24% | 93.40% | 37.68% |
| | QMIX | 0.51 | 0.16 | 0.09 |
| 2v4 scenario | T³OMVP | 0.86 | 0.33 | 0.18 |
| | Improvement | 68.62% | **106.25%** | 100% |

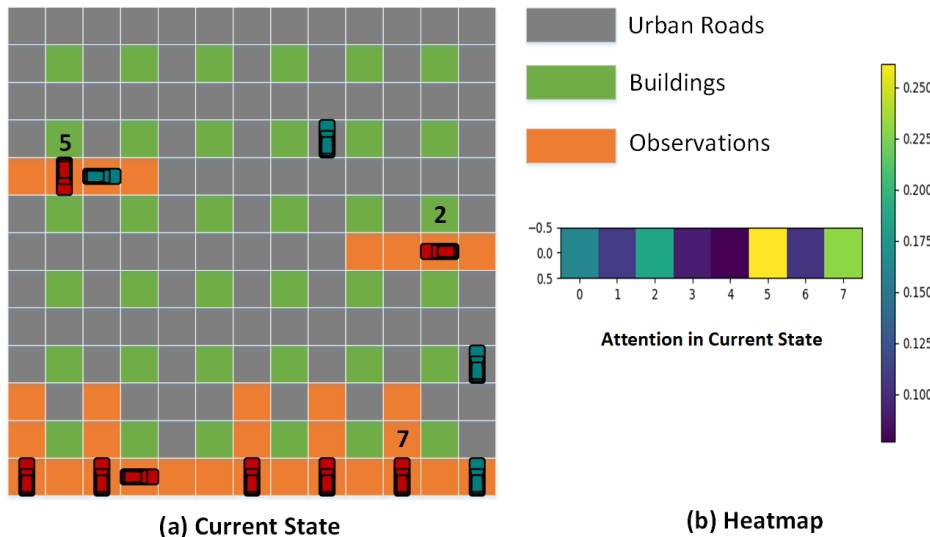

(a) Current State          (b) Heatmap

**Figure 7.** Heatmap of Team Attention. (**a**) shows the current state in which the red agent is the pursuing vehicle, the blue agent is the evading vehicle, the gray area is the road, the yellow area is the observation of the pursuing vehicle, and the green area is the building. (**b**) shows the attention in the current state.

Figure 8 shows the role of hidden states in historical observations. Figure 8a,b show the positional relationship between the pursuing vehicle and the evading vehicle in the previous state and current state, respectively, and Figure 8c shows the attention in the current state and the hidden state retained in the previous state. As shown in Figure 8a,b, because the pursuing vehicle No. 5 did not move, the evading vehicle observed by the pursuing vehicle No. 5 in the previous step drove out of the observation field of the pursuing vehicle No. 5; however, the pursuing vehicle No. 5 still has the highest attention among the attentions in the current state. Due to the hidden state of the previous state, the attention reserved by the pursuing vehicle No. 5 is the highest except for the pursuing vehicle No. 3, because the pursuing vehicle No. 3 does not observe the evading vehicle in both steps and the observation field shrinks due to the complicated road structure; that is why the pursuing vehicle No. 5 can obtain the highest attention at the current state. According to the transmission of the hidden state, the pursuing vehicle can save the experience of historical observations and improve the pursuing efficiency.

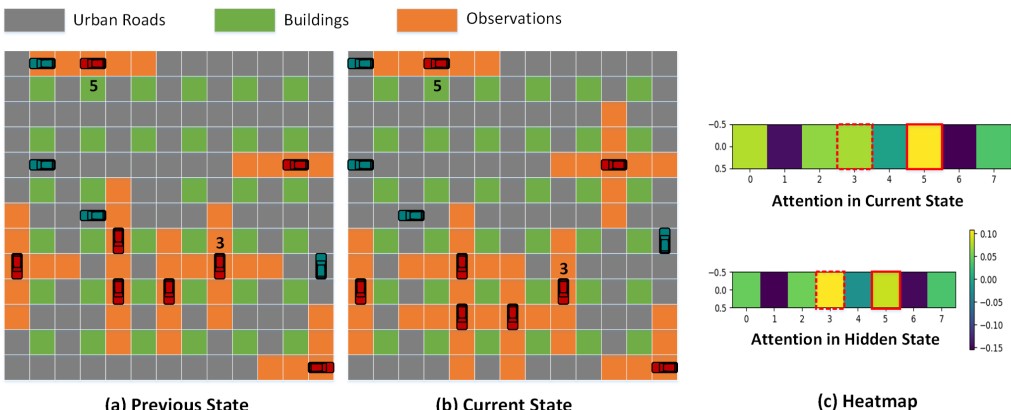

**(a) Previous State**  **(b) Current State**  **(c) Heatmap**

**Figure 8.** Heatmap of Hidden State. (**a**,**b**) show the previous state and current state. The upper of (**c**) shows the attention in current state. The bottom of (**c**) shows the attention in hidden state of previous state.

## 5. Conclusions

An OMVP problem was defined from an MVP problem in which the pursuing vehicles are required to pursue the evading vehicles in the building-blocked environment. A T$^3$OMVP scheme was proposed to solve the OMVP problem in the simulated urban environment which has a complicated road structure, restricted moving spaces, and constrained observations. The T$^3$OMVP scheme integrated a transformer structure into the QMIX to process the time and team's attention to various observations. Different from UPDeT, the T$^3$OMVP used joint observation to collect the observations of all pursuing vehicles instead of the policy decoupling strategy for final attention. The T$^3$OMVP can directly make decisions and achieve a better or the same performance without policy decoupling. This paper modified the predator–prey scenario and obtains a complex urban multi-intersection OMVP simulator, which is a light-weight system with easy to train agents, and the proposed T$^3$OMVP scheme was verified in this simulation environment. More extensive experiments proved that the T$^3$OMVP can achieve a much better performance on more difficult scenarios than the original QMIX and VDN. Through the distribution of attention displayed by the heatmap, it can be seen that adding the transformer structure can coordinate the attention of the team, and the hidden state can store some historical information so that it can be applied to multiple problems based on Dec-POMDPs. Furthermore, according to the performance of the experiments, the T$^3$OMVP can be generalized to more cooperative multi-agent scenarios, such as robot control or power grid dispatching.

For the application of the transformer in reinforcement learning, our future work will focus on the theoretical analysis of transformers, which will help to better explain the application of the attention mechanism in processing joint observation among agents in MARL. Furthermore, the ITS is subject to potential cyberattacks with the increase in connectivity in transportation networks and cyberattacks can inject malicious information into the transportation networks to mislead autonomous vehicles. The other future work will focus on the issue of resilience to cyberattacks in the OMVP for real-world scenarios, which will help us analyze the impact of cyberattacks on the network when exposed from outside. The future work will focus on the modeling and theoretical analysis of transformers. It will greatly help to better explain the application of the attention mechanism in processing joint observation among agents in MARL.

**Author Contributions:** Conceptualization, Z.Y. and L.Z.; methodology, Z.Y. and T.W.; software, Z.Y.; validation, Z.Y., Q.W. and Y.Y.; investigation, Z.Y., Q.W. and Y.Y.; resources, L.Z.; data curation, Z.Y., Q.W. and Y.Y.; writing—original draft preparation, Z.Y.; writing—review and editing, Z.Y., Q.W., Y.Y. and L.L.; visualization, Z.Y.; supervision, L.Z. and L.L.; project administration, Z.Y. All authors have read and agreed to the published version of the manuscript.

**Funding:** This work was partially supported by the National Natural Science Foundation of China (Grant No. 62176024) and project A02B01C01-201916D2.

**Institutional Review Board Statement:** Not applicable.

**Informed Consent Statement:** Not applicable.

**Data Availability Statement:** The code is open-sourced at https://github.com/its-ant-bupt/T3OMVP (last accessed on 27 February 2022).

**Conflicts of Interest:** The authors declare no conflict of interest.

## Abbreviations

The following abbreviations are used in this manuscript:

| | |
|---|---|
| IoVs | Internet of Vehicles |
| MDP | Markov Decision Processes |
| MVP | Multi-Vehicle Pursuit |
| MARL | Multi-agent Reinforcement Learning |
| CTDE | Centralized Training with Decentralized Execution |
| OMVP | Observation-constrained Multi-Vehicle Pursuit |
| T³OMVP | Transformer-based Time and Team Reinforcement Learning Scheme for OMVP |
| Dec-POMDP | Decentralized Partially Observed Markov Decision Processes |

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
