# Peer review of "T3OMVP: A Transformer-Based Time and Team Reinforcement Learning Scheme for Observation-Constrained Multi-Vehicle Pursuit in Urban Area"

_electronics, doi:10.3390/electronics11091339_

Round 1
Reviewer 1 Report
This is a well-written paper on Team Reinforcement Learning Scheme for Observation-constrained Multi-Vehicle Pursuit in Urban Area.
The authors provide a method for Multi-Vehicle Pursuit games
(MVP), a multi-vehicle cooperative ability to capture mobile targets, using
multi-agent systems, multi-agent reinforcement learning, and using internet of vehicles methods.
The presentation is clear, and their method is illustrated by several figures with experimental results.
The equations in the paper are clearly stated and, up to my knowledge are correct. Possibly some more details in the variables of the various equations are welcomed, especially in subsections 33.3.1 & 3.3.2.
In section 4 may be an algorithm– even in pseudocode- that shows their method explicitly could be used.
Ways to improve the paper:
Please refine the abstract section and highlight better the authors’ contribution
The introduction section must be further polished.
Please provide more references on the Reinforcement Learning topic as for the internet of vehicles in the intro.
Accept with minor revisions
Reviewer 2 Report
The topic of the article is important and interesting. There are many individuals and organizations working on this issue. Everyone is looking for a solution. The authors offered their idea and solution to the problem. Mathematical formulas are presented correctly. I would like to ask the authors to give more explanation. So that readers of the magazine who are just starting to work in this area understand them. The charts need to be expanded. It is necessary to write about issues of resilience to cyberattacks. How the network will behave when exposed from outside. The conclusions are correct. I would like to see several generalized work algorithms. Authors should better justify the perspective of their research.
Reviewer 3 Report
This paper proposes a Transformer-based Time and Team Reinforcement Learning scheme to address the problem of multi-vehicle pursuit in the multi-intersection urban area.
The paper is well organized and readable. I have only one suggestion about references.
My personal opinion is that the first three references are added by force. This does not help the journal to increase the impact factor because these references are considered as self-citations of the journal. It has already happened that some journals have lost their impact factor because they have had many more self-citations than pure citations from other journals. I suggest you keep one reference and find the other in the journals with impact factors.
I recommend that the paper should be accepted with minor revision.
Reviewer 4 Report
The conclusions should be detailed.Author Response
Thanks for your comments. Please see the attachment.
